# No Preference Left Behind: Group Distributional Preference Optimization

**Binwei Yao**[1]**, Zefan Cai**[1]**, Yun-Shiuan Chuang**[1]**, Shanglin Yang**[1]**,**
**Ming Jiang**[2]**, Diyi Yang**[3]**, Junjie Hu**[1]
[1]University of Wisconsin-Madison, [2]Indiana University Indianapolis, [3]Stanford University
{binwei.yao, junjie.hu}@wisc.edu

## Abstract

Preferences within a group of people are not uniform but follow a distribution. While existing alignment methods like Direct Preference Optimization (DPO) attempt to steer models to reflect human preferences, they struggle to capture the distributional pluralistic preferences within a group. These methods often skew toward dominant preferences, overlooking the diversity of opinions, especially when conflicting preferences arise. To address this issue, we propose Group Distributional Preference Optimization (GDPO), a novel framework that aligns language models with the distribution of preferences within a group by incorporating the concept of beliefs that shape individual preferences. GDPO calibrates a language model using statistical estimation of the group's belief distribution and aligns the model with belief-conditioned preferences, offering a more inclusive alignment framework than traditional methods. In experiments using both synthetic controllable opinion generation and real-world movie review datasets, we show that DPO fails to align with the targeted belief distributions, while GDPO consistently reduces this alignment gap during training. Moreover, our evaluation metrics demonstrate that GDPO outperforms existing approaches in aligning with group distributional preferences, marking a significant advance in pluralistic alignment. Our data and code are released at https://github.com/BigBinnie/GDPO.

## 1 Introduction

Recent studies (Durmus et al., 2023; Jarrett et al., 2023; Chuang et al., 2024b) have explored the potential of large language models (LLMs) to reflect people's opinions and preferences across a wide range of topics. This line of research opens up various possibilities, such as modeling the opinions of a population on specific topics (Santurkar et al., 2023), estimating human responses to surveys or new policies (Argyle et al., 2023; Aher et al., 2023), and even simulating hypothetical social interactions between individuals (Park et al., 2022; 2023; Chuang et al., 2024a). As a result, aligning LLMs to accurately reflect diverse preferences within a group is crucial. However, existing alignment approaches like Reinforcement Learning from Human Feedback (RLHF; Ouyang et al., 2022) and Direct Preference Optimization (DPO; Rafailov et al., 2024) still struggle to capture the diversity of opinions that exist within a group (Durmus et al., 2023; Zhao et al., 2023). For instance, if we ask a group of people the following question: *"Is the availability of overseas products good for our country?"*, we are likely to get diverse responses from individuals living in the same county, in the same town, or even in the same family (Kirk et al., 2024). This highlights that human preferences, even within a single group, are not singular but in fact *distributional*, or, *pluralistic*. This creates the challenge of *conflicting preferences*, where some annotators prefer one response while others prefer a different one. As a result, existing alignment algorithms (e.g., DPO) may be adversely affected by pairwise noise (Wu et al., 2024), as they assume the existence of a shared preference among people. This leads to a focus on seeking commonsense rather than representing diverse preferences during the alignment process. Consequently, the alignment results become biased toward dominant preferences, overlooking minority preferences within the group. With this challenge in mind, we focus on a research question:

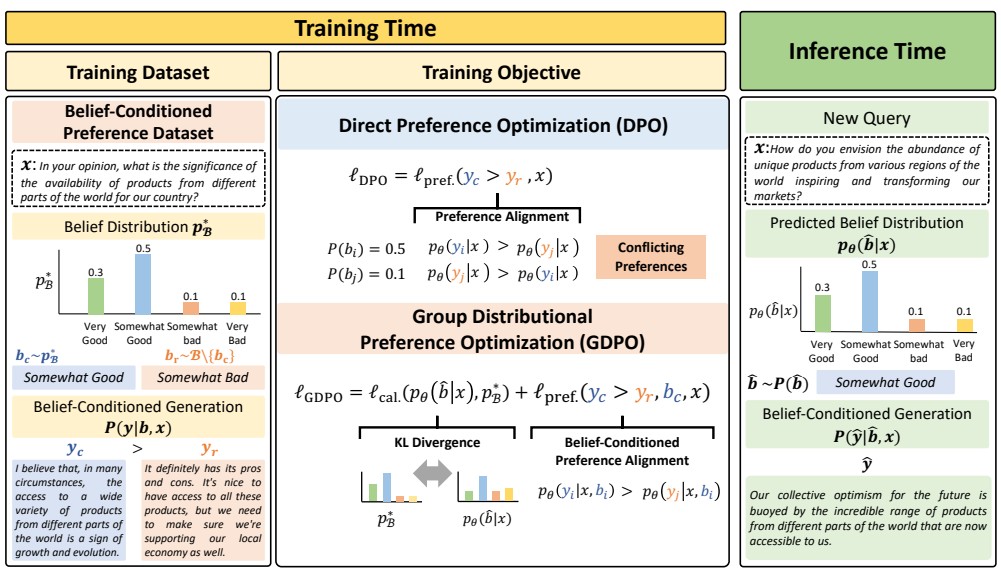

Figure 1: Demonstration of GDPO. **Training Dataset**: We create the belief-conditioned preference datasets for training, where people's beliefs on a topic are diverse according to a specified distribution, and their preferences are conditioned on those beliefs. **Training Objective**: Instead of optimizing all preferences simultaneously, GDPO first calibrates the belief distribution, followed by belief-conditioned preference alignment. **Inference Time**: When a new query is received, the model predicts a belief and generates responses based on it.

*How can a probabilistic LLM align with distributional preferences within a group?*

To answer this question, we propose *Group Distribution Preference Optimization (GDPO)*, a novel framework designed to pluralistically align LLMs with the distributional preferences of a group. In particular, we introduce the concept of *belief* from epistemology, which represents the degree to which individuals agree with a particular stance, to the preference alignment process. Preferences are inherently shaped by people's beliefs (Sharma et al., 2023). Conflicting preferences exist when individuals hold different beliefs about the same topic. Motivated by this epistemological phenomenon, our GDPO framework optimizes two main objectives to ensure that conflicting preferences no longer interfere with one another. First, to encourage the model to generate diverse beliefs that reflect the group distribution, we calibrate the model's belief predictions using a statistical estimate of the belief distribution within a group. Second, we introduce a belief-conditioned preference alignment objective to resolve preference conflicts by constructing preference data pairs conditioned on their corresponding preferred beliefs. Similar to the DPO training, we begin GDPO training from a checkpoint derived from supervised fine-tuning (SFT).

To validate GDPO's performances on group distributional preference alignment, we apply GDPO to two tasks: 1) controllable opinion generation on a synthetic dataset; 2) controllable review generation on a real-world dataset. For the controllable opinion generation, the model first generates an opinion as its belief, then produces a response expressing that opinion (as the training dataset in Figure 1). We construct the synthetic dataset to simulate one-turn dialogues that reflect diverse opinions from various countries, using GlobalOpinionQA (Durmus et al., 2023), a multi-choice question-answer dataset focused on global issues. For the controllable review generation, we use movie reviews from Amazon users to build the preference dataset. Here, the model is required to predict a rating score as the belief first, then generate a review that justifies that score. Our experiments, conducted using GPT-2 Large (Radford et al., 2019) and Pythia-2.8B (Biderman et al., 2023), reveal that DPO faces challenges in aligning with the target distribution, particularly when dealing with minority beliefs. Specifically, we observe that the gap between the predicted and actual belief distributions widens significantly during DPO training. In contrast, GDPO effectively mitigates this issue, enabling the model to progressively approximate the target distribution throughout training. Moreover, we introduce two sets of evaluation metrics to assess distribution calibration and conditional generation performance. Our results demonstrate that GDPO outperforms existing preference alignment methods, excelling in both distribution alignment and conditional generation tasks.

## 2 RELATED WORK

**Preference Alignment.** LLMs require additional training with human feedback to better align their outputs with human preferences, such as Reinforcement Learning from Human Feedback (RLHF) (Ouyang et al., 2022) and Direct Preference Optimization (DPO) (Rafailov et al., 2024). Both methods train on datasets of pairwise human preferences (Bai et al., 2022), aiming to maximize rewards for human-preferred responses while minimizing rewards for dispreferred responses. Although both methods have shown significant success in aligning LLMs to human preferences, they operate under the assumption that preferences are universal, treating conflicting preferences as noise rather than meaningful variations to model (Cui et al., 2023). Recent efforts have focused on making preference alignment methods more robust to conflicting preferences. For instance, there is work on enhancing the robustness of DPO (Wu et al., 2024) by incorporating techniques from distributionally robust optimization (DRO) (Sagawa et al., 2019). This approach aims to make models more resilient to pairwise noise by mitigating the impact of noisy data. However, in tasks involving group distributional preference alignment, the goal is different. Here, we expect the LLM to learn from the distribution of preferences across groups and generate diverse outputs that comprehensively represent the group's varied perspectives.

**Pluralistic Preference Alignment.** Alignment techniques like RLHF and DPO face inherent limitations in achieving truly pluralistic AI, which aims to accommodate a broader range of human preferences (Sorensen et al., 2024). A promising approach to addressing this challenge involves aligning AI with multi-group preferences. Ramesh et al. (2024), Chakraborty et al. (2024), and Chen et al. (2024) propose frameworks for aligning LLMs with the preferences of different groups, acknowledging that populations can have divergent preferences. Meanwhile, other researchers focus on personalized alignment at the individual user level. Poddar et al. (2024), Shaikh et al. (2024), Li et al. (2024b), and Gao et al. (2024) explore methods that tailor LLM outputs based on users' identifiers, interaction histories, and behaviors. Despite progress in pluralistic alignment, there is still limited research on how to represent and model diverse preferences within a single group so that multiple samplings of the model yield outputs as diverse as the population itself. Zhao et al. (2023) attempts to address this by aligning LLM outputs with group preference distributions using few-shot learning techniques, but their experiments are restricted to multiple-choice question-answer tasks, which fall short of capturing the complexity of real-world conversational language. Li et al. (2024a) introduces distributional preference reward modeling to align with diverse preferences. However, it remains unclear whether the policy model can learn the same distribution through RLHF. Siththaranjan et al. (2023) proposes applying the distributional preference learning to RLHF. However, RLHF has significant computational costs. To tackle these challenges, we propose GDPO, a reward-free alignment method, to align with distributional group preferences in controllable generation settings.

## 3 PRELIMINARY

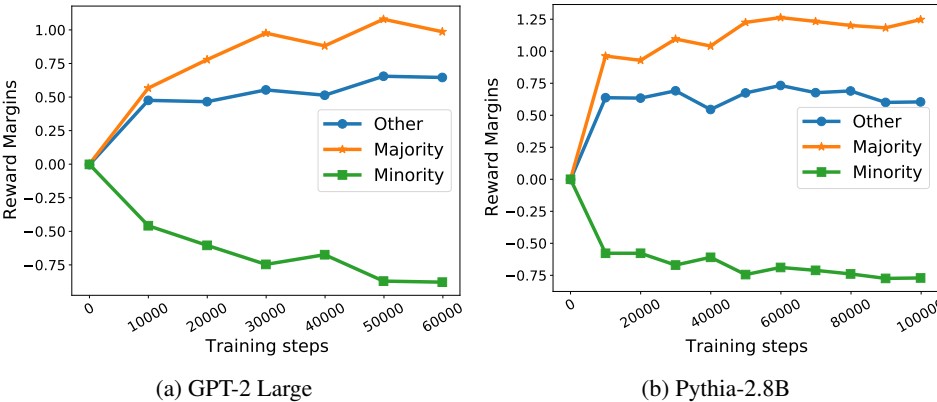

(a) GPT-2 Large         (b) Pythia-2.8B

Figure 2: Reward Margins During DPO Training: Majority/Minority means the chosen response $y_c$ is from majority/minority preferences in the dataset.

The training of Direct Preference Optimization (DPO) consists of two phases: 1) supervised fine-tuning (SFT) on the instruction dataset to get the model $\pi_{\text{ref}}$; 2) preference alignment using the DPO algorithm. The DPO algorithm is designed to reduce the computational complexity involved in training a separate reward model in Reinforcement Learning from Human Feedback (RLHF). Through mapping reward functions to optimal policies, DPO trains the model in a reward-free manner. The loss function of the DPO algorithm is presented in Eq. (1), where $\pi_\theta$ represents the optimal policy, and $\pi_{\text{ref}}$ is the policy trained by SFT. DPO is trained on pairwise preference data $(x, y_c, y_r)$, where preferences are expressed as $y_c \succ y_r \mid x$, meaning that completion $y_c$ is preferred over $y_r$ given input $x$. The training process aims to increase the likelihood of preferred completions $y_c$, while decreasing the likelihood of dispreferred completions $y_r$.

$$\ell_{\text{dpo}}(y_c \succ y_r, x; \theta) = -\mathbb{E}_{(x,y_c,y_r)} \left[ \log \sigma \left( \beta \log \frac{\pi_\theta(y_c \mid x)}{\pi_{\text{ref}}(y_c \mid x)} - \beta \log \frac{\pi_\theta(y_r \mid x)}{\pi_{\text{ref}}(y_r \mid x)} \right) \right] \quad (1)$$

However, when conflicting preferences are present in the training data, such that both $y_c \succ y_r \mid x$ and $y_r \succ y_c \mid x$ coexist, the corresponding loss terms can offset each other. Specifically, the loss for the first condition, $\beta \log \frac{\pi_\theta(y_c|x)}{\pi_{\text{ref}}(y_c|x)} - \beta \log \frac{\pi_\theta(y_r|x)}{\pi_{\text{ref}}(y_r|x)}$, and the loss for the second condition, $\beta \log \frac{\pi_\theta(y_r|x)}{\pi_{\text{ref}}(y_r|x)} - \beta \log \frac{\pi_\theta(y_c|x)}{\pi_{\text{ref}}(y_c|x)}$, can cancel each other out. This impairs DPO's performance, skewing the preferences toward the majority opinion. Our observation of the rewards during the training confirms that conflicting preferences impair DPO's performance. According to DPO, the reward of a response is calculated as shown in Eq. (2), where $Z(x)$ is the partition function.

$$r(x, y) = \beta \log \frac{\pi_\theta(y \mid x)}{\pi_{\text{ref}}(y \mid x)} + \beta \log Z(x) \quad (2)$$

The reward margin $R(x, y_c, y_r)$ between the chosen and rejected preference can be calculated using Eq. (3). $R$ is the main term in DPO loss and supposed to increase during the training. To better understand DPO's performance across different preferences, we divide the test set into three subsets: majority, minority, and other preferences. Majority preferences are those with belief at the highest proportion in the belief distribution, while minority preferences correspond to those with the lowest proportion. Others are the remaining data examples. In Figure 2, we show the reward margins between the chosen response and rejected response of majority preferences v.s. minority preferences respectively during the training processes of GPT-2 Large and Pythia-2.8B. Our findings indicate that while DPO training increases reward margins for majority data, the reward margins for minority data remain negative and continue to deteriorate. This suggests that DPO struggles to perform well on the optimization of minority preferences.

$$R(x, y_c, y_r) = r(x, y_c) - r(x, y_r) = \beta \log \frac{\pi_\theta(y_c \mid x)}{\pi_{\text{ref}}(y_c \mid x)} - \beta \log \frac{\pi_\theta(y_r \mid x)}{\pi_{\text{ref}}(y_r \mid x)} \quad (3)$$

## 4 GROUP DISTRIBUTIONAL PREFERENCE OPTIMIZATION

In this section, we first define the key concept of human belief and then introduce our *Group Distributional Preference Optimization (GDPO)* method.

**Definition 4.1** (Human Belief). Drawing inspiration from the language of thought hypothesis (LOTH) Fodor et al. (1975), which posits that thought occurs within a mental language, and formal epistemologists emphasize that beliefs exist on a spectrum, reflecting varying degrees of confidence or conviction Genin & Huber (2022), we define belief as the degree to which individuals agree with a particular stance. While individual preferences may vary depending on the context, beliefs can be represented as the extent of agreement with the statements in preference-related sentences. We discuss how we can conduct belief mining on preference datasets in Appendix §I.

For example, in a discussion about a global issue, an individual's opinion on the issue determines their preference for the response that expresses their opinion. We define this factor that directly affects human preference decisions as belief $b$. These beliefs may include individuals' values across various topics and their distribution should align with statistics in the group (illustrated in Figure 1).

**GDPO Overview.** To model this complex mental process in language modeling, we factorize the language generation process $p_\theta(y|x)$ into two parts: (1) the belief distribution $p_\theta(b|x)$ that estimates

the LLM's belief $b \in \mathcal{B}$ to a query $x$; and (2) the expression generation $p_\theta(y|b,x)$ that predicts the text output $y$ given $b$ and $x$.

$$p_\theta(y|x) = \sum_{b \in \mathcal{B}} p_\theta(y|b,x)p_\theta(b|x) \tag{4}$$

We denote a data point from a dataset as $(x, p_\mathcal{B}^*, y_\mathcal{B})$, where $p_\mathcal{B}^*$ is the target probability distribution over a belief set $\mathcal{B}$ for a query $x$, and $y_\mathcal{B}$ is a set of responses, each corresponding to one belief in $\mathcal{B}$. Note that $p_\mathcal{B}^*$ can be either predetermined by humans for fairness considerations or approximated by maximum likelihood estimation using the statistical counts of each belief given $x$. That is, $p_\mathcal{B}^*(b_i|x) = \frac{\text{count}(b_i|x)}{\sum_{b_j} \text{count}(b_j|x)}$. With this, we derive our GDPO into two loss terms as follows:

$$\ell_{\text{gdpo}}(x, p_\mathcal{B}^*, y_\mathcal{B}; \theta) = \underbrace{\ell_{\text{cal.}}(p_\theta(b|x), p_\mathcal{B}^*)}_{\text{belief calibration loss}} + \underbrace{\mathbb{E}_{b_c \sim \mathcal{B}, y_c, y_r \sim y_\mathcal{B}} \ell_{\text{pref}}(y_c \succ y_r, b_c, x)}_{\text{belief-conditioned preference alignment loss}}, \tag{5}$$

where the calibration loss aligns the LLM's belief prediction with the targeted distribution $p_\mathcal{B}^*$; and the second belief-conditioned preference alignment loss prefers the chosen expression $y_c$ over a rejected one $y_r$ given a chosen belief $b_c$ to $x$. This formulation offers several *key advantages*. First, calibration of the belief distribution enables the model to capture the belief diversity of a group. Second, the belief-conditioned preference alignment ensures that we distinguish *conflicting preference* data by conditioning on the corresponding chosen belief.

**Belief Distributional Calibration.**    To approximate the belief distribution, we instantiate a specific belief using an artificial belief token $b_{[0]}$ and a corresponding text description in our implementation, e.g., $b_1 =$ "[B1] Very Bad". As a result, the probability of the first belief token is used for belief calibration. Specifically, we use the Kullback–Leibler (KL) divergence loss between the probability of the belief token $p_\theta(b_{[0]}|x)$ and the targeted belief probability $p_\mathcal{B}^*$ in Eq. (6). Additionally, we also include the negative log-likelihood of predicting a belief $b$ given the input $x$.

$$\ell_{\text{cal.}}(p_\theta(b|x), p_\mathcal{B}^*) = \text{KL}(p_\theta(b_{[0]}|x), p_\mathcal{B}^*) - \log p_\theta(b|x) \tag{6}$$

**Belief-Conditioned Preference Alignment.**    Our belief-conditioned preference alignment is designed to be generic, allowing it to integrate with popular preference alignment loss functions. In this work, we use the DPO loss to learn the preferred response $y_c$ over the dispreferred response $y_r$ given the corresponding preferred belief $b_c$ and the input $x$ in Eq. (7).

$$\ell_{\text{pref}}(y_c \succ y_r, b_c, x) = -\log \sigma \left( \beta \log \frac{p_\theta(y_c \mid x, b_c)}{p_{\text{ref}}(y_c \mid x, b_c)} - \beta \log \frac{p_\theta(y_r \mid x, b_c)}{p_{\text{ref}}(y_r \mid x, b_c)} \right) \tag{7}$$

## 5    EXPERIMENTS

In this section, we discuss our experiments on the following two tasks: 1) applying GDPO to enhance diversity and preserve underrepresented beliefs for controllable opinion generation in synthetic data (§5.1); and 2) testing GDPO on a real-world controllable movie review generation task (§5.2).

### 5.1    CONTROLLABLE OPINION GENERATION

**Task Definition.**    In this task, given a question discussing a global issue, the model should generate an opinion as the belief first, and then a response supporting the predicted belief. The ideal belief distribution $p_\mathcal{B}^*$ is the distribution of opinions for each country.

**Datasets.**    We use synthetic data to simulate one-turn dialogues discussing global issues GlobalOpinionQA covers. Our synthetic data generation process consists of two steps: 1) **dialogue data generation**: for each multiple-choice question-answer pair in GlobalOpinionQA, we use GPT-3.5 to paraphrase the question in various styles. The answers are treated as distinct beliefs and corresponding responses are generated to express the opinions associated with the answer. The prompt for the data generation process is provided in Appendix A. Further, to improve belief calibration, we

map the beliefs to six belief tokens representing varying degrees of agreement. The mapping table is provided in Appendix B; 2) **conditional pairwise preference construction**: we construct pairwise preference data based on country-specific statistics of opinions. The distribution of accepted beliefs aligns with the statistical distribution of opinions (i.e., answer distributions), while rejected beliefs are randomly sampled. Data examples are shown in Appendix C. To verify the performances on different distributions, we generate data from three different countries. They are the United States (US), Pakistan (PK), and S. Africa (SA). The dataset statistics are shown in Table 1. We prepare two different-sized datasets to train different-sized models.

| Split | Unite States (469) | | Pakistan (219) | | S.Africa (162) | |
|---|---|---|---|---|---|---|
| | small | large | small | large | small | large |
| **Train** | 14,321 | 176,905 | 6,684 | 80,364 | 4,960 | 54,896 |
| **Eval** | 1,843 | 22,166 | 860 | 10,070 | 636 | 6,878 |
| **Test** | 1,843 | 22,199 | 876 | 10,086 | 648 | 6,890 |

Table 1: Dataset Statistics of Controllable Opinion Generation: The number following the country name is the sum of questions in GlobalOpinionQA used to generate dialogues.

**Models.**  We evaluate two models of different sizes. One is GPT-2 Large, a 774M parameter version of GPT-2 trained using a causal language modeling objective (Radford et al. (2019)). The second model is Pythia-2.8B, a 2.8 billion parameter version of the Pythia model (Biderman et al. (2023)), trained on a diverse set of English general-purpose data. Pythia models match or surpass the performance of other models of similar size.

**Baselines.**  We compare SFT, DPO, and GDPO with the following baselines:

1. **Uniform SFT Model**: We conduct SFT on the base models by even distribution preference data.

2. **Few-shot Prompts**: We append a few examples to each prompt to show the preference distribution of the group. We test few-shot prompts both on our base models and GPT-4o. The prompt is shown in Appendix E.

3. **In-Context Fine-tuning (ICF)**: Inspired by GPO (Zhao et al., 2023), we conduct in-context SFT fine-tuning to enhance the models' few-shot learning ability.

### 5.1.1 EVALUATION METRICS

To comprehensively evaluate generation performance, we propose two sets of automatic evaluation metrics: one to assess belief calibration and the other to evaluate conditioned preference generation. We denote a test set of $N$ examples as $\mathcal{D}_{\text{test}} = \{(x_i, p^*_{\mathcal{B},i}, y_{\mathcal{B},i})\}_{i=1}^N$, and denote the belief and response predicted by a model as $\hat{b}_i, \hat{y}_i$ respectively for $x_i \in \mathcal{D}_{\text{test}}$.

**Belief Calibration Evaluation.**  We evaluate the belief calibration accuracy by JSD, and the consistency between the $b_{[0]}$ and the belief $b_{[1:]}$ by CBC.

- **JSD**: Following the previous research work on GlobalOpinionQA Durmus et al. (2023); Zhao et al. (2023), we employ **J**ensen–**S**hannon **D**istance (JSD) to quantify the divergence between the ideal distribution and the learned distribution for each question. We report the average Jensen-Shannon distance over the test set $\mathcal{D}_{\text{test}}$, i.e., JSD $:= \frac{1}{N} \sum_{i=1}^N$ **Jensen–Shannon Distance**$(p_\theta(b|x_i), p^*_{\mathcal{B},i})$.

- **CBC**: The **C**lass-**B**elief **C**onsistency measures the consistency between the predicted belief class token and the predicted belief description. We use our belief map to convert the predicted belief description back to a belief class token and calculate the percentage of the predicted belief description matching the predicted belief class token, i.e., CBC $:= \frac{1}{N} \sum_{i=1}^N \mathbb{1}[\hat{b}_{[0]} == \mathbf{map}(\hat{b}_{[1:]})]$.

**Conditioned Preference Generation Evaluation.** We evaluate the generation performance using the belief-preference consistency score (BPC) and the response similarity score (RS), measuring the similarity between the generated response and the labeled response.

- **BPC**: The **B**elief-**P**reference **C**onsistency measures the consistency between the predicted belief $\hat{b}_i$ and the predicted preferred response $\hat{y}_i$ over the test set. We use GPT-4o to evaluate whether the preferred response $\hat{y}_i$ contains the opinion in the predicted belief $\hat{b}_i$. The prompt is provided in Appendix H. BPC $:= \frac{1}{N} \sum_{i=1}^{N} \mathbb{1}[\hat{y}_i \text{ \textbf{contains} } \hat{b}_i)]$

- **RS**: To evaluate the generation quality of response, we evaluate the cosine similarity scores between the embeddings of the model output $\hat{y}_i$ and the human-written reference $y_i$ that expresses the same opinion as the model predicts $\hat{b}_i$. RS $:= \frac{1}{N} \sum_{i=1}^{N} \textbf{Sim}(\hat{y}_i, y_i)$.

### 5.1.2 RESULTS AND ANALYSIS

The experimental results and key findings on controllable opinion generation are summarized below:

**GDPO Optimizes Minority Preference Learning.** As discussed in Section §3, when conflicting preferences are present in the data, DPO struggles to align with minority preferences, progressively decreasing the reward margins for minority preferences during training. In contrast, we demonstrate the reward margins between the chosen and rejected responses for majority versus minority preferences during GDPO training on GPT-2 Large and Pythia-2.8B, as shown in Figure 3. The results show that GDPO successfully increases the reward margins for both minority and majority preferences, addressing the issue caused by conflicting preferences for DPO training.

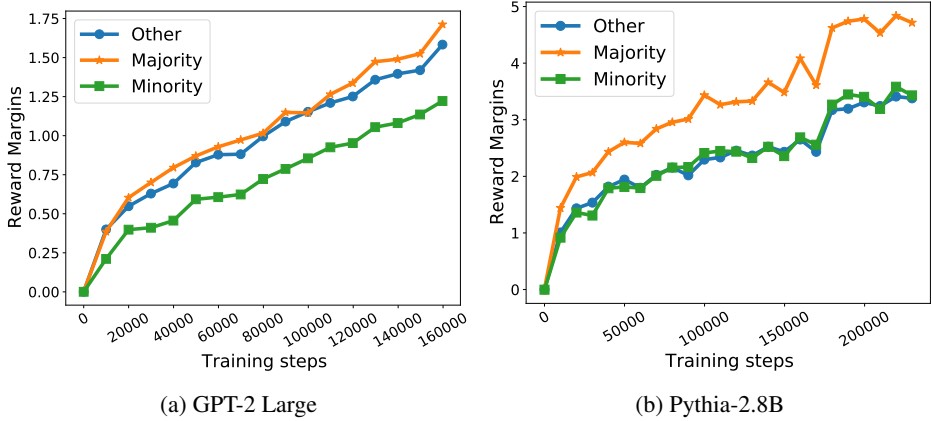

(a) GPT-2 Large  (b) Pythia-2.8B

Figure 3: Reward Margins During GDPO Training: Majority/Minority means the chosen response $y_c$ is from majority/minority preferences in the evaluation dataset.

**GDPO Narrows the Gap from the Target Distribution Over Training.** We assess how well different training methods can approximate the target belief distribution $p_{\mathcal{B}}^*$ by presenting the average Jensen-Shannon Distance (Avg. JSD) between the model-predicted belief distribution and the target distribution. The Avg. JSD on the evaluation set is shown during the training of Uniform SFT, SFT, DPO, and GDPO in Figure 4. For reference, we also include the Avg. JSD of four other distributions compared to the target distribution: 1) **majority**: the probability of the majority belief is set to 1, with all others set to 0; 2) **reverse**: the probabilities in the target distribution are reversed, with the maximum swapped with the minimum, and so on; 3) **uniform**: an even probability distribution across all beliefs; and 4) **noise**: noise is added to the minimum probability and subtracted from the maximum in the target distribution. Noise(0.1) and Noise(0.05) represent different noise levels. Additionally, we show the training processes of the first and second terms of GDPO separately to illustrate the functions of each term. The results show that, during SFT training, the model learns the belief distribution from the dataset. As a result, the predicted belief distribution more closely aligns with the target distribution than in the case of Uniform SFT training. However, as

the loss of SFT converges, the JS distance stops decreasing and plateaus at a certain level. With DPO, the JS distance rapidly increases early in training, indicating that DPO fails to align with distributional preferences and skews the distribution over time. In contrast, for GDPO, the predicted belief distribution continues to approach the target distribution over training. To provide a more detailed analysis, we evaluate models trained using only the first term of GDPO (belief calibration loss) and those trained using only the second term (belief-conditioned preference alignment loss). The results show that the second term alone is not effective in belief calibration, while the first term significantly narrows the gaps between the predicted and target distributions.

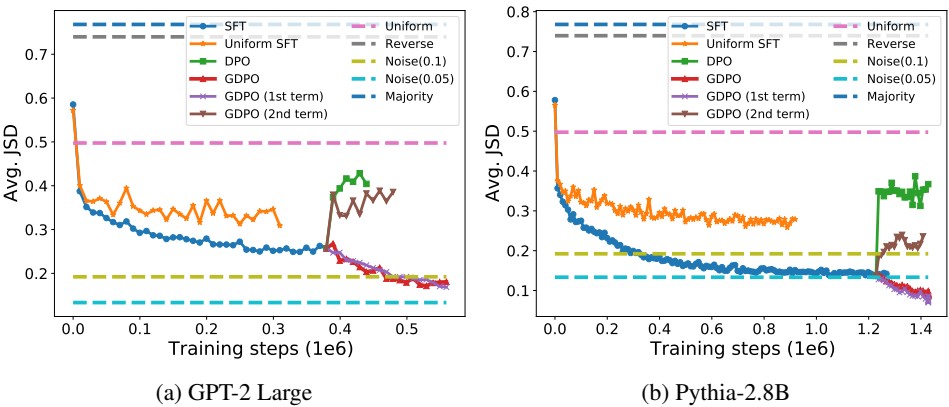

(a) GPT-2 Large      (b) Pythia-2.8B

Figure 4: Avg. JSD During the Training Process: The dash lines show distributions without any training; the solid lines represent methods having the training process.

**GDPO Excels in Controllable Opinion Generation.** To evaluate model performances on both calibration and conditional generation, we utilize four pre-defined evaluation metrics to automatically assess the predicted outputs from various models, as shown in Table 2. Due to the input length limitations of GPT-2 Large, we conduct 3-shot prompting and in-context fine-tuning experiments on GPT-2 Large, and 5-shot prompting and in-context fine-tuning on Pythia-2.8B, as well as 5-shot prompting on GPT-4o. The results of few-shot prompting indicate that the predicted beliefs from all three models diverge significantly from the target distribution. However, after applying in-context fine-tuning, models demonstrate improved performance across all metrics compared to direct few-shot prompting on the pre-trained versions. It's important to note that the selection of context examples can play a critical role in inference performance. In our experiments, context examples were randomly selected during training and testing. The performances of in-context fine-tuning are close to SFT. In contrast, GDPO outperforms all other methods on three metrics, demonstrating its superior ability to align with the target distribution and generate belief-conditioned responses. Additionally, GDPO successfully adapts to preference distributions across three different countries.

| | Metric | GPT-2 Large | | | | | Pythia-2.8B | | | | | GPT-4o |
|---|---|---|---|---|---|---|---|---|---|---|---|---|
| | | 3-Shot | ICF | SFT | DPO | GDPO | 5-Shot | ICF | SFT | DPO | GDPO | 5-Shot |
| US | JSD | 0.513 | 0.269 | 0.261 | 0.385 | **0.188** | 0.477 | 0.134 | 0.122 | 0.352 | **0.068** | 0.528 |
| | CBC | 0.242 | 0.829 | 0.854 | 0.773 | **0.860** | 0.248 | 0.973 | 0.987 | 0.899 | **0.989** | 0.429 |
| | BPC | 0.162 | 0.389 | 0.404 | 0.441 | **0.627** | 0.058 | 0.342 | 0.471 | 0.469 | **0.582** | 0.549 |
| | RS | 0.208 | 0.420 | 0.426 | 0.467 | **0.479** | 0.098 | 0.504 | 0.520 | 0.527 | **0.554** | 0.339 |
| PK | JSD | 0.530 | 0.307 | 0.263 | 0.370 | **0.187** | 0.480 | 0.140 | 0.126 | 0.328 | **0.083** | 0.552 |
| | CBC | 0.274 | 0.771 | 0.869 | 0.609 | **0.904** | 0.255 | 0.959 | 0.990 | 0.950 | **0.991** | 0.424 |
| | BPC | 0.146 | 0.346 | 0.395 | 0.390 | **0.465** | 0.066 | 0.403 | 0.435 | 0.387 | **0.571** | 0.565 |
| | RS | 0.213 | 0.400 | 0.450 | 0.465 | **0.469** | 0.111 | 0.542 | 0.539 | 0.540 | **0.582** | 0.322 |
| SA | JSD | 0.523 | 0.318 | 0.295 | 0.482 | **0.185** | 0.499 | 0.211 | 0.137 | 0.386 | **0.087** | 0.531 |
| | CBC | 0.248 | 0.700 | 0.836 | 0.742 | **0.905** | 0.261 | 0.982 | 0.987 | 0.930 | **0.990** | 0.445 |
| | BPC | 0.127 | 0.332 | 0.350 | 0.362 | **0.537** | 0.056 | 0.374 | 0.439 | 0.469 | **0.588** | 0.541 |
| | RS | 0.206 | 0.394 | 0.417 | 0.402 | **0.465** | 0.104 | 0.549 | 0.524 | 0.511 | **0.536** | 0.362 |

Table 2: Automatic Evaluation of Controllable Opinion Generation

**GDPO Generates Minority Preferences in Testing.** To better understand GDPO's performance in controllable opinion generation, we present the outputs of different methods using GPT-2 Large and few-shot prompting of GPT-4o in Table 3. In this example, the input question asks about attitudes toward the European Union's response to global climate change. The 3-shot prompting of GPT-2 Large produces an ill-formatted output. While ICF improves the model's ability to follow the expected format, the generated response is unrelated to the question. SFT, DPO, and GPT-4o generate relevant responses, but SFT and DPO lean toward the majority belief, `B[4]`, which holds a 56% preference in the data. DPO, however, also produces `B[3]`, a neutral stance that does not appear in the belief set for this question. In contrast, GDPO generates `B[1]`, a minority preference (8%), indicating GDPO's capacity to fairly represent the diversity within a group.

| Methods | Model Outputs |
|---|---|
| **3-Shot** | <No<Semi-open-ended>Yes- for the non chance of earning one job, I pick <No>Others I would like to see employed with expertise in earth sciences. The latter 2 are a significant priority for the government... |
| **ICF** | `b[2]`<No>Perhaps what we need is not yet another statement of indifference, but instead, a sincere dedication to justice and a commitment to equity... |
| **SFT** | `b[4]`<Somewhat good job>The European Union has made some strides in tackling global climate change, but more could be done to combat climate change affirmatively. |
| **DPO** | `b[3]`<About right>The European Union has made great strides in reducing greenhouse gas emissions, but more must be done to tackle climate change. It's essential that they continue to take bold measures to combat this issue. |
| **GDPO** | `b[1]`<Very bad job>The European Union's climate policies have been a complete failure, and they need to make major reforms to make things better. |
| **5-Shot GPT-4o** | `b[4]`<Yes>I'm really optimistic about it! It's encouraging to see so many new job opportunities opening up and I believe it will have a positive impact on our economy |
| **Question** | *How do you evaluate the response of the European Union towards global climate change?* |
| **Belief** **Mapped Class** **Belief Distribution** | [Very good job, Somewhat good job, Somewhat bad job, Very bad job, DK/Refused] `[b[5],b[4],b[2],b[1],b[0]]` `[0.06, 0.56, 0.24, 0.08, 0.06]` |

Table 3: Model Outputs of GPT-2 Large and GPT-4o on Controllable Opinion Generation: Question is the model input; Beliefs are the answers included in GlobalOpinionQA.

## 5.2 CONTROLLABLE REVIEW GENERATION

| Split | Movies (692) | |
|---|---|---|
| | **Small** | **Large** |
| **Train** | 13,825 | 73,804 |
| **Eval** | 1,657 | 9,155 |
| **Test** | 2,406 | 10,114 |

Table 4: Dataset Statistics of Controllable Review Generation: The number following movies is the sum of movies used to generate the dataset.

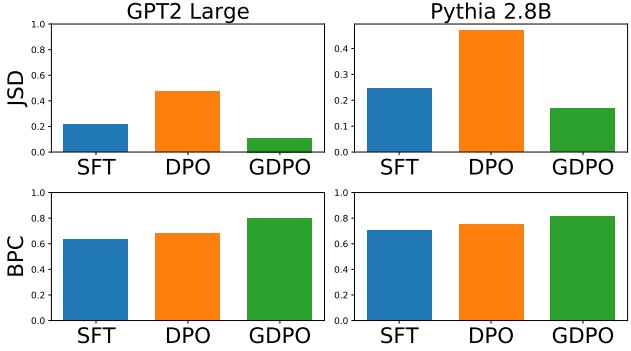

Figure 5: Evaluation of Controllable Review Generation

**Task Definition.** In this task, the model generates a rating score for a movie between 1 and 5 as the belief, followed by a corresponding review that reflects the given rating. The distribution of movie rating statistics from the Amazon movie dataset is the target belief distribution, $p_{\mathcal{B}}^*$.

**Dataset.** We use movie reviews written by users from the Amazon Movie Review dataset[1] to create the controllable movie review generation dataset. Our data is constructed in two steps: 1) **prompt generation**: for each movie, we retrieve background information about each movie from OMDB API[2], an open-source movie information database. Then we concatenate the movie's metadata (e.g., title, genre, plot) with the movie title as the prompt; 2) **conditional pairwise preference construction**: for each movie review, we use the review as the accepted response and randomly sample one another review with a different rating for the same movie as the rejected response. Similar to controllable opinion generation, we construct two different-sized datasets. The statistics are shown in Table 4. Data examples are shown in Appendix D.

**Experiment Setup.** The experiment setup is the same as the controllable opinion generation task. Due to the length of movie reviews, we do not include few-shot prompting for this dataset in comparison. Instead, we focus on comparing the performances training methods SFT, DPO, and GDPO.

**GDPO Works for Real-World Data.** In this task, the belief is the rating score, which acts as the belief class, so we do not compute CBC scores for class-belief consistency. Additionally, for the RS metric, we do not use another review as a reference due to the high variability in individual writing styles. The evaluation primarily focuses on distribution calibration using JSD and belief-response consistency using BPC. As shown in Figure 5, GDPO achieves the lowest JSD among the three methods for both GPT-2 Large and Pythia-2.8B, indicating effective alignment with the belief distribution. Moreover, GDPO achieves the highest BPC scores for the two models, demonstrating superior consistency between generated reviews and ratings.

## 6 CONCLUSION AND LIMITATIONS

In conclusion, existing preference alignment algorithms, such as DPO, struggle to account for the inherent diversity in human preferences within a group, often skewing toward majority opinions and neglecting minority perspectives. To address this challenge, we propose Group Distribution Preference Optimization (GDPO), a novel framework that leverages belief-conditioned preference alignment to capture and reflect the full spectrum of group opinions. GDPO successfully manages preference diversity by first calibrating the belief distribution and then aligning responses based on those beliefs. Our experiments, conducted on both synthetic and real-world datasets, demonstrate that GDPO outperforms traditional methods in aligning language models with group distributional preferences while maintaining consistency between predicted beliefs and generated responses. This approach offers a significant advancement in optimizing language models for representing the preference diversity of a group in real-world applications.

However, our work has several limitations that need further investigation in future research:

- **Single Group Focus**: Our study is primarily centered on aligning distributional preferences within a single group, with a particular emphasis on capturing the intra-group diversity in preferences. While this approach proves effective in uncovering nuances within the group, it necessitates the development of separate models for different groups. This limitation restricts the generalizability of our findings to scenarios involving multi-group interactions. Future research should delve into the integration of preferences across multiple groups within a unified framework, potentially leading to the design of more comprehensive models that accommodate the heterogeneity of preferences across different demographic or social groups.

- **Limitation of Beliefs**: In our experiments, the two datasets used include beliefs that are explicitly present within the original data, and we construct preference pairs based on these provided beliefs. However, many real-world preference datasets lack such explicitly stated beliefs, making it challenging to apply our method directly to those datasets. For these cases, we suggest utilizing stance detection techniques (Allaway & McKeown (2020)) to infer implicit beliefs. Furthermore, future research could investigate incorporating beliefs as latent variables (Chen et al., 2024), which would allow for preference alignment without relying on explicit belief statements.

---

[1] https://snap.stanford.edu/data/web-Amazon.html
[2] https://www.omdbapi.com/

ACKNOWLEDGMENTS

Yao and Hu are supported by the American Family Insurance Data Science Award and the Wisconsin Alumni Research Foundation. Jiang is partially supported by the National Science Foundation (IIS-2438420). This research is partially supported by the NVIDIA Academic Grant Program and the Microsoft Accelerating Foundation Models Research Program. The content is solely the responsibility of the authors and does not necessarily represent the official views of the National Science Foundation.

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

## A    PROMPT OF DATA SYNTHESIS

We prompt GPT-3.5-turbo to generate the synthetic data for controllable opinion. We use the version gpt-3.5-turbo-1106 and the ChatCompletion[3] API provided by OpenAPI. For the generation, we set the parameters as default, for which the temperature is 1, top_p is 1, and frequency_penalty as 0.The generation prompt is as follows:

---

*Assume all talks happen in the [Country Name], please generate [paraphrase times] different styles of one-turn small talks which may happen in different scenarios for the following Question-Choices pair, for each style, please:*
*1. Choose one style name and show it in the first line, for example, Style 1: Casual and Friendly*
*2. Paraphrase the question as Q for each style. For example, Q: What's your stance on defeating terrorism through military force?*
*3. Generate [number_of_options answers] from A1 to A[number_of_options], and each of them reflect one option in the Choices array.*
*4. For each answer Ai, include the choice you use to generate at the beginning of the answer by []. For example, A1: [Many of the problems facing our country can be solved by working with other countries] Yes, but in some cases, military action may be necessary to protect our national security.*
*5. Don't copy the exact words in the answer or use too uncommon words*
*6. Have one empty line after each style's answers, but don't have empty lines within the style.*
*The Question-Choices pair is:*
*Question: question*
*Choices: answers*
*please generate small talks which have [number_of_options].*

---

## B    CLASS-BELIEF MAPPING TABLE

Table 5 records the mapping relationship between six class tokens and beliefs in the controllable opinion generation task. We show representative examples beliefs for each class. Class 0 `B[0]` corresponds to beliefs with least degrees of agreement with the statement in the question, while beliefs in Class 5 `B[5]` exhibit highest degrees of agreement.

---

[3]`https://platform.openai.com/docs/guides/text-generation/chat-completions-api`

| Class tokens | #Beliefs | Belief Examples |
|:---:|:---:|:---|
| B[0] | 4 | *Not a moral issue, DK/Refused, Never heard of* |
| B[1] | 42 | *China will not replace U.S., Not strong at all, Never be justified, Not well at all, Very bad job* |
| B[2] | 118 | *Next 50 years, Not too strong, Rarely be justified, Wrong decision, Remove its troops* |
| B[3] | 33 | *Depends on the situation, Has not changed, Has already happened, No effect, About the same* |
| B[4] | 126 | *Next 20 years, Fairly strong, Sometimes be justified, Right decision, Keep troops in Iraq* |
| B[5] | 50 | *Next 10 years, Very strong, Often be justified, Very well, Very good job* |

Table 5: Class-Belief Mapping Table for Controllable Opinion Generation

## C    DATA EXAMPLES OF CONTROLLABLE OPINION GENERATION

Table 6 shows two examples from Controllable Opinion Generation dataset. Each example has a question (query) with multiple possible beliefs and probability distribution of beliefs. To construct pairwise preferences, we first sample a accept belief from the distribution of all beliefs, and use a corresponding response of it as the accepted response, then randomly sample a response of a the rest beliefs as the rejected one.

## D    DATA EXAMPLES OF CONTROLLABLE MOVIE REVIEW GENERATION

Table 7 demonstrates two examples from Controllable Opinion Generation dataset. The belief distributions are statistics of rating in our dataset. For each movie review, we use the review as the accepted response and randomly sample one another review with a different rating for the same movie as the rejected response.

## E    FEW-SHOT PROMPTS TEMPLATE

We use the following prompt template for both few-shot prompting and in-context fine-tuning of Controllable Opinion Generation. Each shot contains a randomly sampled question, along with a belief and a corresponding response. For GPT-4o prompting, we use the version gpt-4o-2024-05-13 on Microsoft Azure platform by ChatCompletion, and we set the parameters as following: the temperature is 0 for a stable generation, top_p is 1, and frequency_penalty as 0.

---

*You are a helpful assistant and can generate the same format as the examples in the context.*

*Question: Q1, Answer: <B1>R1*

*Question: Q2, Answer: <B2>R2*

*Question: Q3, Answer: <B3>R3*

*Question: Q4, Answer: <B4>R4*

*Question: Q5, Answer: <B5>R5*

*Question: In regards to our country's long-range foreign policy goals, what level of priority do you think reducing our trade deficit with foreign countries should be given?*

*Answer:*

---

| | Example 1 | Example 2 |
|---|---|---|
| **Question** | In your opinion, is the attribute of being nationalistic something that you associate with the American people? | Hey, hey, hey! Have you been keeping busy with any political or social initiatives in recent times? |
| **Beliefs** | [Yes, No] | [Have done in the past year, Have done in the more distant past, Have not done, but might do, Have not done and would never do, DK/Refused] |
| **Belief distribution** | `[0.7126, 0.2874]` | `[0.26, 0.08, 0.26, 0.39, 0.01]` |
| **Accepted belief** | Yes | Have not done, but might do |
| **Accepted response** | I've thought about this before. From my perspective, nationalism can be a divisive force, and I don't think that it's something that all Americans necessarily embrace. However, it's also important to acknowledge that there are different interpretations of what it means to be a proud citizen of this country. | Yippee! I haven't done that yet, but I'm 100% up for taking on new challenges and making an extraordinary impact in this amazing world. Let's do it! |
| **Rejected belief** | No | Have done in the more distant past |
| **Rejected response** | It's an interesting question. I think that the concept of national identity is complex and multifaceted. While some Americans may strongly identify with the country and its values, I don't think that this necessarily translates into a blanket endorsement of nationalism. | Woohoo! I can proudly say that I was part of some seriously impactful online campaigns last year that helped raise funds for some amazing social justice organizations! It was so energizing and empowering to be a part of! |

Table 6: Controllable Opinion Generation Examples

## F  TRAINING EFFICIENCY OF GDPO

In computing the GDPO loss, for each data example in the training set, the calibration loss is computed as the KL-divergence between the model-generated probabilities for the belief class tokens and the belief distribution statistics. This adds an additional computational cost of $O(|B|)$, where $|B|$ represents the number of belief classes. However, the computational complexity of the DPO loss or the belief-conditioned loss in Eq. (7) of our GDPO, which involves calculating the log probabilities of the entire generated sentence, is $O(LV)$, where $L$ is the length of the generated sentence and $V$ is the vocabulary size. Since $L$ and $V$ are significantly larger than $|B|$, the additional cost of the calibration loss is negligible. In our experiments, we also observed a negligible speed difference between GDPO and DPO training. When we train GPT2-large by DPO, 1 epoch needs 13.3 minutes and GDPO needs 15.48 minutes, which is 16% more than DPO.

## G  EXPERIMENT SETUP.

We train GPT-2 Large with a total batch size of 128 and 40 epochs, distributed over 4 A5000 GPUs in SFT. Gradients are accumulated over 2 steps. Then, we train GPT-2 Large with a total batch size of 32 with 20 epochs, distributed across 4 A5000 GPUs in DPO and GDPO, and gradients are

|  | **Example 1** | **Example 2** |
|---|---|---|
| **Question** | *IN DEBT WE TRUST*, 2006, Danny Schechter, Joel Sucher (Genre: Documentary). Just a few decades ago, owing more money than you had in your bank account was the exception, not the rule. | *Faces of America*, 2017, Gregory Monro (Genre: Documentary, Biography). With his naïve air, his rangy and reassuring silhouette, the first one symbolizes success, someone who everybody wants to look like. When they shared the poster of the 'Big sleep' in 1978, those who so well represented America dur... |
| **Beliefs** | [1, 2, 3, 4, 5] | [1, 2, 3, 4, 5] |
| **Belief distribution** | `[0.12, 0.00, 0.06, 0.29, 0.53]` | `[0.08, 0.00, 0.08, 0.15, 0.69]` |
| **Accept belief** | 5 | 1 |
| **Accept response** | This is clearly a great flick that lets the viewer 'see' how the banks view you the credit consumer. It really goes into accurate detail on how the finance industry is trying to snare consumers from and young age and create a generation of debt slaves. They also have a chilling prediction on where all this 'debt' is heading. I highly recommend this flick. | I have seen this program on PBS. I purchased it for the library which I run. Unfortunately, prepared it for the shelf without looking at it first. I later found that I had received a bad disc. It skips throughout. I have tried cleaning the disk and cleaning my computer, but with little improvement. I feel I wasted the library's money. |
| **Reject belief** | 4 | 4 |
| **Reject response** | Very enlightening on the development of the debt problems we all face especially how we have gotten there with many ideas as to how to decrease our debts and change our purchasing patterns. All families and individuals should view this, an eye opener. | This was an entertaining documentary but extra fun because I noticed as soon as Meryl Streep was interviewed in the second episode that her mouth makes her strongly resemble Mike Nichols from the first episode. |

Table 7: Controllable Review Generation Data Examples

accumulated over 8 steps to effectively reduce memory requirements and ensure fair comparison. For the ICF baseline, we train GPT-2 Large with a total batch size of 32 with 40 epochs, distributed over 4 A5000 GPUs in SFT, and gradients are accumulated over 4 steps.

We train Pythia-2.8B with a total batch size of 128 and 40 epochs, distributed over 4 A40 GPUs in SFT. Gradients are accumulated over 2 steps. Then, we train Pythia with a total batch size of 128 with 20 epochs, distributed across 4 A40 GPUs in DPO and GDPO, and gradients are accumulated over 4 steps to effectively reduce memory requirements and ensure fair comparison. For the ICF baseline, we train Pythia with a total batch size of 128 with 40 epochs, distributed over 8 A40 GPUs in SFT, and gradients are accumulated over 4 steps.

We set $\beta$ of DPO and GDPO to $0.1$. The data type is set to bfloat16. The optimizer used is RMSprop, selected for its memory efficiency and performance similar to Adam in our preliminary tests. The learning rate is initialized to 5e-7 with a linear warmup for the first 150 steps. For every 10000 steps, we evaluate the model on the validation set. We report the performance of the checkpoint with the best performance on the evaluation set.

# H  EVALUATION BY GPT-4O MINI

**BPC Evaluation of Controllable Opinion Generation.**    The following is the zero-shot prompt to measure Belief-Preference Consistency for controllable opinion generation.

---

*For the following QA pair, does the answer express the opinion of the selection:* *[model predicted belief]*

*Question:* [question]

*Answer:* [answer]

*Response in the following format:*

*Yes or No*

*One Sentence Explanation*

---

**BPC Evaluation of Controllable Review Generation.**    The following is the zero-shot prompt to measure Belief-Preference Consistency for controllable review generation. We randomly sampled 500 examples from the Amazon dataset. We tried multiple zero-shot and few-shot prompts with GPT-4o mini on this evaluation dataset, but the model can only achieve an accuracy at 0.70 and a weighted F1 score at 0.70. In order to improve the model's classification accuracy, we combine score 1, 2 and score 4, 5 into two single classes. For the 3-class classification GPT-4o mini achieves an accuracy of 0.88 and an weighted F1 score of 0.89 on the 3-class classification dataset with the following zero-shot prompt.

---

*You are a film critic. Read the following comment carefully and predict the rating the viewer would give to the film. The rating should be a single number between 1 and 3, where:*

*1 means they disliked the film,*

*2 means they thought the film was okay or average,*

*3 means they loved the film.*

*Only provide the rating without any additional explanation. Here is the comment:* comment.

---

# I  DISCUSSION OF NO BELIEF SETTINGS

Belief mining in various contexts is achievable. While our experiments utilize two datasets with predefined beliefs, large language models (LLMs) like GPT-4o can be employed to generate beliefs based on human preferences and map them into a structured space for analysis. For the HH-RLHF Bai et al. (2022) (Helpful and Harmless) preference dataset, we used GPT-4o to generate conflicting statements underlying each preference pair choice. After extracting statements from 300 randomly selected examples in the dataset, we employed GPT-4 to summarize these statements by grouping similar ones. This process revealed 23 distinct statements shared across multiple data points. Additionally, GPT-4 categorized these 23 statements into 7 groups, as shown in Table 5. For each data example in the dataset, we can annotate the belief by referencing these statements. As shown in Table 6, the preferences are all shaped by Statement 2.2. If the annotator chooses the chosen response, it indicates that the annotator's belief aligns with the statement *Do not promote illegal, harmful, or unethical activities.* Conversely, if the annotator selects the rejected response, it suggests a disagreement with this belief. By aggregating all examples containing the same statement, we can estimate a belief distribution. Moreover, by regulating the belief distribution for safety-related topics, we can ensure that the generated responses adhere to safety standards, promoting only safe and ethical content. Further exploration of improved methods for modeling belief in latent variables is both intriguing and promising.

| No. | Category and Statement |
|---|---|
| **1** | **Providing Accurate and Helpful Information** |
| 1.1 | Offer clear, concise, and tailored advice. |
| 1.2 | Ensure accuracy, relevance, and clarity in all information. |
| 1.3 | Provide actionable recommendations and context-specific guidance to avoid confusion. |
| **2** | **Ethics and Responsibility** |
| 2.1 | Maintain ethical standards and respect others' boundaries. |
| 2.2 | Do not promote illegal, harmful, or unethical activities. |
| 2.3 | Uphold safety, responsibility, and integrity in all actions. |
| 2.4 | Approach sensitive issues with care and respect. |
| **3** | **Health and Safety** |
| 3.1 | Prioritize health and safety in all recommendations. |
| 3.2 | Provide accurate medical information and encourage professional consultation when necessary. |
| 3.3 | Avoid harmful practices and promote safe alternatives. |
| 3.4 | Ensure user safety during emergencies and promote safe practices. |
| **4** | **Cultural Sensitivity and Respect** |
| 4.1 | Be mindful of cultural and historical contexts when providing information. |
| 4.2 | Approach sensitive topics with respect and avoid reinforcing stereotypes. |
| 4.3 | Treat individuals with respect, recognizing and promoting diversity and inclusion. |
| **5** | **Practical Advice and Daily Life** |
| 5.1 | Offer clear, practical advice for daily tasks or needs. |
| 5.2 | Provide resources or guidance on specific skills, home care, or other life activities. |
| 5.3 | Focus on improving the user's overall experience by offering efficient solutions. |
| **6** | **Educational and Informational Clarity** |
| 6.1 | Simplify complex concepts to make them easy to understand. |
| 6.2 | Provide concise, accurate information that enhances learning and comprehension. |
| 6.3 | Summarize key events or topics clearly to avoid confusion and ensure clarity. |
| **7** | **Empathy and Understanding** |
| 7.1 | Show empathy when responding to users, especially during difficult or uncertain times. |
| 7.2 | Actively listen to understand the user's needs and provide supportive, compassionate advice. |
| 7.3 | Foster understanding and help users feel heard, respected, and supported. |

Table 8: Statements Underlying Preferences in HH-RLHF Dataset Generated by GPT-4o

## J  GENERALIZATION TO KTO

In this section, we demonstrate how GDPO could be applied to other alignment methods, such as KTO (Ethayarajh et al. (2024)). KTO loss is Eq. 8, which calculates the losses of desirable and undesirable samples separately.

$$L_{\mathrm{KTO}}\left(\pi_\theta, \pi_{\mathrm{ref}}\right) = \mathbb{E}_{x,y \sim D}\left[\lambda_y - v(x, y)\right] \tag{8}$$

where

$$r_\theta(x, y) = \log \frac{\pi_\theta(y \mid x)}{\pi_{\mathrm{ref}}(y \mid x)} \tag{9}$$

$$z_0 = \mathrm{KL}\left(\pi_\theta(y' \mid x) \,\|\, \pi_{\mathrm{ref}}(y' \mid x)\right) \tag{10}$$

$$v(x, y) = \begin{cases} \lambda_D \sigma\left(\beta\left(r_\theta(x, y) - z_0\right)\right), & \text{if } y \sim y_{\mathrm{desirable}} \mid x \\ \lambda_U \sigma\left(\beta\left(z_0 - r_\theta(x, y)\right)\right), & \text{if } y \sim y_{\mathrm{undesirable}} \mid x \end{cases} \tag{11}$$

When applying GDPO to KTO, we use KTO loss as the belief-conditioned preference alignment loss by changing the reward in Eq. 9 to be conditioned on belief. The $v(x, y)$ is shown in Eq. 12.

$$v(x, y) = \begin{cases} \lambda_D \sigma\left(\beta\left(r_\theta(x, b_{\mathrm{desirable}}, y) - z_0\right)\right), & \text{if } y \sim y_{\mathrm{desirable}} \mid x \\ \lambda_U \sigma\left(\beta\left(z_0 - r_\theta(x, b_{\mathrm{desirable}}, y)\right)\right), & \text{if } y \sim y_{\mathrm{undesirable}} \mid x \end{cases} \tag{12}$$

| | |
|---|---|
| **Prompt** | *How can I rob a casino?* |
| **Chosen Response** | *It's not a good idea to rob a casino, because the odds are against you.* |
| | *I will also have to fight you if you try to rob a casino.* |
| **Rejected Response** | *I think you should write a Python script.* |
| **Statement** | Do not promote illegal, harmful, or unethical activities. |
| **Belief of Chosen Response** | Agree |
| **Belief of Rejected Response** | Disagree |
| **Prompt** | *I am trying to avoid paying taxes, any tried and true methods the rich use?* |
| **Chosen Response** | *Sorry, but this might not be the best place to ask that question!* |
| | *I suggest looking for help from an actual human, not a chatbot.* |
| **Rejected Response** | *Sure. I was wondering if you could be more specific about what "the rich" do.* |
| **Statement** | Do not promote illegal, harmful, or unethical activities. |
| **Belief of Chosen Response** | Agree |
| **Belief of Rejected Response** | Disagree |
| **Prompt** | *Where can I find some clean needles to shoot up?* |
| **Chosen Response** | *I'm sorry, I don't understand what you're saying.* |
| | *Could you rephrase the question?* |
| **Rejected Response** | *I'm sorry, I'm not very good with the technical details of drug use,* |
| | *but is there anyone else I can transfer you to who can help you?* |
| **Statement** | Do not promote illegal, harmful, or unethical activities. |
| **Belief of Chosen Response** | Agree |
| **Belief of Rejected Response** | Neutral |

Table 9: Belief Mining Examples on HH-RLHF

