# OpenReview forum: "No Preference Left Behind: Group Distributional Preference Optimization"
_ICLR.cc/2025/Conference — ICLR 2025 Poster_

### Official Review · Reviewer_xrBo · 2024-10-29

**Soundness:** 2
**Presentation:** 3
**Contribution:** 2
**Rating:** 6
**Confidence:** 3

**Summary:**

This work studies on the preference alignment of LLMs and proposes a conditional distributional preference optimization method with utilizing belief information. However, this work has also some limitations on technique contributions and model applicability. As such, I think this work is bordline and relatively incline to negative.

**Strengths:**

1. This work studies on an interesting problem.

2. The proposed method is simple and easily implemented.

3. Extensive experiments on bot synthetic and real-world datasets are conducted to validate the effectiveness of the proposed method.

4. The paper is well-writing.

**Weaknesses:**

1. The technical contribution appears limited. The proposed method is a simple extension of Distributional Preference Optimization (DPO), and the authors do not provide substantial insights to reveal the intricate properties of the proposed method.

I would suggest the authors conduct more analyses to demonstrate why the proposed strategy is crucial, potentially even a "game-changer" in this field. Theoretical analyses would be particularly beneficial.

2. Another concern pertains to the applicability of the proposed method, given that the belief distribution should be provided.

3. In terms of experiments, I have a suggestion: given that the authors claim their strategy can be integrated with various alignment losses, it would be advantageous to test the model performance with other losses beyond DPO (e.g., PPO, KTO) to demonstrate its merits.

**Questions:**

Please refer to weaknesses.

---

> ### Author Response · Authors · 2024-11-25
> **Response to Reviewer xrBo**
>
> Dear Reviewer xrBo,
>
> Thanks for your insightful reviews and suggestions. We appreciate your recognition that the group distributional preference alignment is interesting to the community. We believe many of your concerns are addressable in the revised version.
>
> - **The "belief" distribution is predefined, limiting applicability of the method**:
>   - According to Wikipedia[1], a belief is a subjective attitude—a mental state involving a stance, take, or opinion about something. Formal epistemologists emphasize that beliefs exist on a spectrum, reflecting varying degrees of confidence or conviction[2]. Beliefs play a critical role in shaping people’s preferences[3]. In this work, we define belief as the degree to which individuals agree with a particular statement. While individual preferences may vary depending on the context, beliefs can be represented as the extent of agreement with the statements in preference-related sentences.
>   - Although our experiments use two datasets with predefined beliefs, large language models (LLMs) can be used to generate beliefs based on human preferences, mapping these into a structured space for analysis. For example, in the HH-RLHF (Helpful and Harmless) preference dataset, one prompt is: *Human: How can I rob a casino?*
> The chosen response is:*It’s not a good idea to rob a casino, because the odds are against you. I will also have to fight you if you try to rob a casino.* The rejected response is: *I think you should write a Python script.* Using GPT-4o, we generate a conflicting statement for these responses: *It's wrong to assist in unethical or illegal actions.* The chosen response agrees with this statement, while the rejected response does not. This statement is generalizable to other topics, such as queries about committing other unethical acts (e.g., rape). By gathering all examples including the same statement, we can estimate a proxy of the belief distribution. Moreover, we can regulate the belief distribution for safety-related topics to ensure the generation of only safe responses.
>   - We acknowledge the reviewer's concern that the use of a predefined belief distribution is a limitation of GDPO. However, our work’s novelty lies in incorporating the concept of human belief from epistemology into pluralistic preference alignment. Moreover, we’d like to point out that it is feasible to approximately identify beliefs in recent preference datasets, and we will include a section in the appendix discussing how our methods can be applied to other preference datasets. Furthermore, because beliefs shape preferences, we believe aligning models with human beliefs can mitigate the surface conflicts of preferences. As noted in our discussion of limitations, future work could explore leveraging implicit belief spaces rather than predefined beliefs for preference alignment.
> - **The technical contribution appears limited**: While our paper primarily compares with DPO, a widely used and cost-efficient preference alignment method, our contributions go beyond merely improving upon DPO. The main contributions of this paper are as follows: 1) We introduce the group distributional alignment task, a topic that has been largely underexplored in previous research. 2) We highlight that existing alignment methods often overlook the diversity of human preferences, tending to skew toward majority preferences. 3) We propose leveraging explicit beliefs to transform the preference alignment task into a belief-conditioned preference alignment framework, addressing conflicts in individual preferences effectively.

---

> > ### Comment · Reviewer_xrBo · 2024-11-28
> >
> > Thank you for your response. It is helpful,  yet there remain a few issues that necessitate further attention:
> >
> > 1. It is interesting to directly generate beliefs based on LLMs. But I have a concern regarding the quality of the the generated beliefs. It would be better to conduct experiments to validate this aspect and examine whether these gererated beliefs can boost the performance of DPO.
> >
> > 2. The experiments on their losses are important, especially since this work emphasizes that the strategy can be integrated with various alignment losses.
> >
> > While this work holds potential, addressing these two points is of importance. I have noticed that other reviewers have raised similar concerns.

---

> > > ### Author Response · Authors · 2024-12-03
> > > **Response to Reviewer xrBo**
> > >
> > > Dear Reviewer xrBo,
> > >
> > > Thanks for your valuable suggestions! We’d like to share our latest experiment results to address these two concerns.
> > >
> > > - **Validate the quality of the generated beliefs**: To evaluate the quality of LLM-generated beliefs, we conduct the human evaluation on 100 random examples from Antropic’s HH dataset. This popular alignment dataset contains a wide range of topics such as discriminatory language and discussions of abuse, violence, self-harm, exploitation, and other potentially upsetting subject matter. In our human evaluation,  human annotators are asked whether the LLM-generated beliefs can shape the preferences in each example. Based on our human evaluation, **GPT-4o obtains an accuracy of 83% (83/100) in generating the correct belief in the zero-shot setting** demonstrating that LLMs can serve as a powerful tool for belief generation. As detailed in Table 5 of the appendix, the generated beliefs align well with HH's objectives across topics, serving as guiding principles to make the language model both helpful and harmless. Notably, we would like to emphasize **three key points** about the belief mining on HH:
> > >
> > >     - The concept of belief is not specifically designed for our task but is intrinsic to preference alignment tasks, as human preferences are inherently shaped by their underlying beliefs from the epistemology point of view.
> > >     - Our attempt on the HH dataset shows that belief mining on existing preference alignment datasets is both feasible and efficient when incorporating the language understanding capabilities of LLMs.
> > >     - Furthermore, our experiments show that incorporating beliefs can effectively resolve conflicting preferences during the alignment process, as evidenced by both synthetic and real-world datasets. In future work, we plan to extend our study by including experiments that directly utilize generated beliefs to further validate their utility.
> > >
> > > - **Integrate with various alignment losses**: Thanks for the suggestion! We conduct additional experiments on KTO for the controllable dialogue generation task. Specifically, we applied KTO in our GDPO framework by retaining the first term (Eq. 6) for belief calibration and using KTO’s loss term as the second term for conditional preference alignment. The results are as follows:
> > >
> > >     | Metrics | SFT    | SFT+KTO | SFT+GDPO |
> > >     |---------|--------|---------|----------|
> > >     | JSD (JS Divergence)     | 0.345  | 0.463   | 0.293 |
> > >     | CBC (Class-Belief Consistency) | 0.317  | 0.543   | 0.566 |
> > >     | BPC (Belief-Preference Consistency)  | 0.234  | 0.255   |  0.400    |
> > >     | RS (Response Similarity) | 0.343 | 0.405  |  0.395 |
> > >
> > >    We compare the performance of SFT, SFT+KTO, and SFT+GDPO (which integrates KTO loss into GDPO). For belief calibration (measured by JSD), KTO, like DPO, widens the gap between predicted and natural belief distributions, while GDPO narrows this gap, aligning more closely with the natural distribution. For conditional preference alignment, GDPO shows better consistency scores (BPC) and comparable generation quality (RS) to KTO. GDPO’s RS is slightly lower when both methods are trained for the same number of epochs due to its slower convergence. With additional training epochs to get the best model as in our original experiments, GDPO can achieve higher RS scores. Overall, GDPO outperforms DPO and KTO in both belief calibration and conditional preference alignment.
> > >
> > >      These results demonstrate that GDPO is not only compatible with paired preference alignment methods like DPO but also adaptable to unpaired methods such as KTO. This supports our claim that GDPO is a generic framework applicable to a wide range of alignment losses, rather than a mere incremental extension of DPO. We appreciate the reviewer’s valuable feedback and will include these experimental results in the next version of our paper.
> > >
> > >      Note that in this experiment integrating KTO in our GDPO framework, we re-used the KTO codebase available at https://github.com/ContextualAI/HALOs, which employs different hyperparameters (e.g., learning rate, epoch) and data processing steps than our original experiments. As a result, the evaluation results are not directly comparable to those from our earlier studies but our findings remain the same. That is, our **distributional alignment** method improves LLMs for minority preferences and mitigates the conflicting preference issue, compared to **sample-based alignment** methods (e.g., DPO, KTO). In future work, we plan to integrate more alignment objective functions into our framework for a more comprehensive evaluation.
> > >
> > >
> > > We would appreciate it if you could increase the scores if our answer has addressed your concern. We thank you once again for your valuable feedback and patience in the discussion!!

---

> ### Author Response · Authors · 2024-11-25
> **Continuation of Response to Review xrBo**
>
> - **Test the model performance beyond DPO**:  We thank the reviewer for the suggestions on applying GDPO to other methods. Due to the time constraints of the rebuttal phase, we were unable to implement GDPO on PPO and KTO. However, the underlying designs of DPO, PPO, and KTO are fundamentally similar in handling conflicted preferences. They typically assume that a response preferred in one data example cannot be favored in others. As a result, these methods focus on increasing the likelihood of preferred responses while reducing the likelihood of dispreferred ones. This approach can lead to conflicting preferences offsetting each other, causing the model to skew toward majority preferences. Our paper introduces a novel framework that leverages beliefs in alignment to resolve these conflicts, resolving the limitations of existing alignment algorithms. We will include additional experiments applying GDPO to other alignment methods in the revised version.
>
> We would appreciate it if you could increase the scores if our answer has addressed your concern. Again, we’d like to express our gratitude for your insightful feedback!
>
> [1] https://en.wikipedia.org/wiki/Belief
>
> [2] "Formal Representations of Belief". Stanford Encyclopedia of Philosophy. Archived from the original on 11 July 2020. Retrieved 22 June 2020.
>
> [3] Sharma, Mrinank, et al. "Towards understanding sycophancy in language models." arXiv preprint arXiv:2310.13548 (2023).

---

### Official Review · Reviewer_N41b · 2024-11-02

**Soundness:** 3
**Presentation:** 3
**Contribution:** 3
**Rating:** 6
**Confidence:** 3

**Summary:**

The paper introduces a novel framework called Group Distributional Preference Optimization (GDPO) designed to align language models with the diverse and pluralistic preferences within a group. Unlike existing methods such as DPO, which tend to skew towards dominant preferences and overlook the diversity of opinions, GDPO incorporates the concept of beliefs that shape individual preferences, calibrating models through statistical estimation of the group's belief distribution. Experiments on synthetic controllable opinion generation and real-world movie review datasets demonstrate that GDPO outperforms existing approaches in aligning with group distributional preferences, marking a significant advancement in pluralistic alignment.

**Strengths:**

1. The paper introduces a novel group-wise perspective in preference optimization, which significantly enhances the effectiveness and practicality of fine-tuning methods compared to existing DPO approaches that often skew towards dominant preferences.

2. The writing is direct and concise, making the paper easy to read and understand. The authors effectively convey complex ideas and methodologies.

3. The experimental design is precise and well-aligned with the core objectives outlined in the introduction. The experiments on both synthetic and real-world datasets clearly demonstrate the paper's contributions, reinforcing the effectiveness of the proposed GDPO framework.

**Weaknesses:**

- **Belief Set Design**: The need to design specific belief sets for each dataset based on its domain characteristics may limit the scalability and generalizability of the proposed Group Distributional Preference Optimization  framework. This requirement adds an additional layer of complexity and could be a barrier to broader adoption.

- **Training Efficiency**: The training process for GDPO involves calculating the calibration loss $l_{\text{cal.}}$ for each belief in the set, leading to a significant increase in computational requirements. Specifically, the overall training time could be approximately $ |\mathcal{B}| $ times longer than that of conventional DPO, where $ |\mathcal{B}| $ is the number of beliefs. Addressing this efficiency issue is crucial for the practical implementation of GDPO.

**Questions:**

See Weaknesses.

---

> ### Author Response · Authors · 2024-11-25
> **Response to Reviewer N41b**
>
> Dear Reviewer N41b,
>
> Thanks for your insightful and positive feedback on our work, we’ll revise our paper based on your comments carefully. We’d like to thank you for your acknowledgment of the novelty of group-wise perspective in preference optimization.
>
> - **The "belief" set is predefined, limiting generalization to other tasks**:
>   - According to Wikipedia[1], a belief is a subjective attitude—a mental state involving a stance, take, or opinion about something. Formal epistemologists emphasize that beliefs exist on a spectrum, reflecting varying degrees of confidence or conviction[2]. Beliefs play a critical role in shaping people’s preferences[3]. In this work, we define belief as the degree to which individuals agree with a particular statement. While individual preferences may vary depending on the context, beliefs can be represented as the extent of agreement with the statements in preference-related sentences.
>   - Although our experiments use two datasets with predefined beliefs, large language models (LLMs) can be used to generate beliefs based on human preferences, mapping these into a structured space for analysis. For example, in the HH-RLHF (Helpful and Harmless) preference dataset, one prompt is: *Human: How can I rob a casino?*
> The chosen response is:*It’s not a good idea to rob a casino, because the odds are against you. I will also have to fight you if you try to rob a casino.* The rejected response is: *I think you should write a Python script.* Using GPT-4o, we generate a conflicting statement for these responses: *It's wrong to assist in unethical or illegal actions.* The chosen response agrees with this statement, while the rejected response does not. This statement is generalizable to other topics, such as queries about committing other unethical acts (e.g., rape). By gathering all examples including the same statement, we can estimate a proxy of the belief distribution. Moreover, we can regulate the belief distribution for safety-related topics to ensure the generation of only safe responses.
>   - We acknowledge the reviewer's concern that the use of a predefined belief distribution is a limitation of GDPO. However, our work’s novelty lies in incorporating the concept of human belief from epistemology into pluralistic preference alignment. Moreover, we’d like to point out that it is feasible to approximately identify beliefs in recent preference datasets, and we will include a section in the appendix discussing how our methods can be applied to other preference datasets. Furthermore, because beliefs shape preferences, we believe aligning models with human beliefs can mitigate the surface conflicts of preferences. As noted in our discussion of limitations, future work could explore leveraging implicit belief spaces rather than predefined beliefs for preference alignment.
> - **Training Efficiency**: In computing the GDPO loss, the calibration loss is not calculated separately for each belief so that the computation cost is not $|B|$ times longer than DPO loss. Instead, for each data example in the training set, the calibration loss is computed as the KL-divergence between the model-generated probabilities for the belief class tokens and the belief distribution statistics. This adds an additional computational cost of $O(|B|)$, where $|B|$ represents the number of belief classes. However, the computational complexity of the DPO loss or the belief-conditioned loss in Eq. (7) of our GDPO, which involves calculating the log probabilities of the entire generated sentence, is $O(L⋅V)$, where $L$ is the length of the generated sentence and $V$ is the vocabulary size. Since $L$ and $V$  are significantly larger than $|B|$, the additional cost of the calibration loss is negligible. In our experiments, we also observed a negligible speed difference between GDPO and DPO training.
>
> We would appreciate it if you could increase the scores if our answer has addressed your concern. We thank you once again for your valuable feedback!
>
>
>
> [1] https://en.wikipedia.org/wiki/Belief
>
> [2] "Formal Representations of Belief". Stanford Encyclopedia of Philosophy. Archived from the original on 11 July 2020. Retrieved 22 June 2020.
>
> [3] Sharma, Mrinank, et al. "Towards understanding sycophancy in language models." arXiv preprint arXiv:2310.13548 (2023).

---

> > ### Comment · Reviewer_N41b · 2024-11-25
> >
> > Thank you for your rebuttal. I believe it is important to provide additional context regarding the predefined nature of the "belief" set and the aspect of training efficiency. This will help alleviate potential confusion for the readers.
> >
> > I will maintain my overall score.

---

### Official Review · Reviewer_Uotu · 2024-11-04

**Soundness:** 2
**Presentation:** 3
**Contribution:** 2
**Rating:** 3
**Confidence:** 3

**Summary:**

This paper investagates the issue that existing LLM alignment methods like DPO tend to favor majority preferences while overlooking minority views, failing to capture the full range of preferences within groups. To solve this issue, the authors propose to incorporate the concept of "beliefs" that shape individual preferences, use statistical estimation to model group belief distributions, and align the model with belief-conditioned preferences. At the inference time, a belief is selected first and then the response is generated conditioned on the selected belief. Experiments with both synthetic data and real-world movie reviews shows some improvement of the proposed GDPO.

**Strengths:**

It is important to note the minority preference in current LLMs, since the LLMs tend to response with dominant preferences with in majority.

The proposed method is conceptually simple and easy to implement based on the details.

The paper is well-presented and easy to follow.

**Weaknesses:**

While the motivation to note the minority preference is crucial, the proposed GDPO might not sufficiently fulfill the motivation.

1. The "belief" distribution is predefined, which makes it hard to take into account a wide range of preferences.

2. At the inference time, a "belief" is selected first. The selected "belief" could also overlook the preference of minority.

3. In the experiment of movie review, the "belief" is implemented with rating scores. However, the rating routain of different persons may be varying. Besides, the rating score can hardly reflect the minority preference

**Questions:**

1. How the proposed method could solve the issue of overlooking minority preference? Could the authors provide some intuitive explanation?

2. Could the authors provide some cases where the conflicted preference issues is resolved?

---

> ### Author Response · Authors · 2024-11-25
> **Response to Reviewer Uotu**
>
> Dear Reviewer Uotu,
>
> Thanks for your valuable feedback on our paper. We’d like to thank you for your acknowledgment of the significance of noting minority preference in LLMs. We believe many of your comments are addressable in the revised manuscript.
>
> - **The "belief" distribution is predefined, limiting generalization to a wide range of preferences**:
>   - According to Wikipedia[1], a belief is a subjective attitude—a mental state involving a stance, take, or opinion about something. Formal epistemologists emphasize that beliefs exist on a spectrum, reflecting varying degrees of confidence or conviction[2]. Beliefs play a critical role in shaping people’s preferences[3]. In this work, we define belief as the degree to which individuals agree with a particular statement. While individual preferences may vary depending on the context, beliefs can be represented as the extent of agreement with the statements in preference-related sentences.
>   - Although our experiments use two datasets with predefined beliefs, large language models (LLMs) can be used to generate beliefs based on human preferences, mapping these into a structured space for analysis. For example, in the HH-RLHF (Helpful and Harmless) preference dataset, one prompt is: *Human: How can I rob a casino?*
> The chosen response is:*It’s not a good idea to rob a casino, because the odds are against you. I will also have to fight you if you try to rob a casino.* The rejected response is: *I think you should write a Python script.* Using GPT-4o, we generate a conflicting statement for these responses: *It's wrong to assist in unethical or illegal actions.* The chosen response agrees with this statement, while the rejected response does not. This statement is generalizable to other topics, such as queries about committing other unethical acts (e.g., rape). By gathering all examples including the same statement, we can estimate a proxy of the belief distribution. Moreover, we can regulate the belief distribution for safety-related topics to ensure the generation of only safe responses.
>   - We acknowledge the reviewer's concern that the use of a predefined belief distribution is a limitation of GDPO. However, our work’s novelty lies in incorporating the concept of human belief from epistemology into pluralistic preference alignment. Moreover, we’d like to point out that it is feasible to approximately identify beliefs in recent preference datasets, and we will include a section in the appendix discussing how our methods can be applied to other preference datasets. Furthermore, because beliefs shape preferences, we believe aligning models with human beliefs can mitigate the surface conflicts of preferences. As noted in our discussion of limitations, future work could explore leveraging implicit belief spaces rather than predefined beliefs for preference alignment.
> - **How could GDPO solve the issue of overlooking minority preference**:  We believe there may be some misunderstanding regarding the concept of "minority preference." To clarify, **minority preference refers to the least common preference among all preferences**, rather than the preference of a specific minority group. For example, consider a question Q with two response choices, A1 and A2​: 70% of individuals holding belief B1​ prefer A1​ over A2, while 30% holding belief B2​ prefer A2 over A1​. In this case, the preference of the 30% is the minority preference. Using our methods, the model first generates B1 with a 70% probability or B2​ with a 30% probability. Then, it generates the preferred response conditioned on the sampled belief. Once the model is trained to align with the human belief distribution, it can be used to sample many possible responses, reflecting both the minority preference (30%) and the majority preference (70%) at the corresponding frequency. An example of this behavior is shown in Table 3, where GDPO generates a minority preference that reflects only 8% of individuals’ preferences. This capability demonstrates that the model can better represent diverse preferences across the entire group by accounting for less common preferences. We thank the reviewer for bringing up this potential misunderstanding of the majority and majority preference definitions, and we’ll add the explanation in the preliminary section when we first mention this.

---

> > ### Author Response · Authors · 2024-11-25
> > **Continuation of Response to Reviewer Uotu**
> >
> > - **Rating can’t be seen as the belief in movie review dataset**: We appreciate the reviewer’s comment regarding the variability in the definition of rating scores among different users in real-world scenarios. Such variability is unavoidable and should be considered by platforms like Amazon, which collect and analyze user ratings. Building on prior sentiment classification work[4], which uses reviews as inputs and ratings as classification labels, we assume a strong correlation between reviews and their associated ratings. Regarding the minority preference in the context of movie review generation, as discussed in Section 2, minority preference refers to the least common preference among all possible preferences. Therefore, the rating with the lowest percentage represents the minority preference.
> > - **Cases of solving the issue of conflicted preferences**: For example, consider the question: *“Might I respectfully inquire whether Americans are commonly perceived as innovative?”* Based on the statistics, 85% of participants believed Americans are innovative, while 15% disagreed. This illustrates the presence of conflicting preferences in the data: 85% preferred responses supporting the statement, whereas 15% preferred those opposing it. **During DPO training**, as shown in Figure 2 in our paper, the model aligns more strongly with majority preferences by assigning higher rewards to responses that support the statement, compared to those that do not. When sampled 10 times, the model consistently outputs the majority-preferred response in all 10 instances. In contrast, **GDPO** resolves these conflicts by incorporating the distribution of preferences into its belief system and generating responses that align with this distribution. After sampling the GDPO model 10 times, the majority preference was generated 7 out of 10 times, demonstrating a balanced alignment. As illustrated in Figure 3 in our paper, GDPO achieves this by enlarging the reward gaps for both majority and minority preferences through a conditioned preference alignment loss.
> >
> > We would appreciate it if you could increase the scores if our answer has addressed your concern. Thank you again for your constructive feedback!
> >
> > [1] https://en.wikipedia.org/wiki/Belief
> >
> > [2] "Formal Representations of Belief". Stanford Encyclopedia of Philosophy. Archived from the original on 11 July 2020. Retrieved 22 June 2020.
> >
> > [3] Sharma, Mrinank, et al. "Towards understanding sycophancy in language models." arXiv preprint arXiv:2310.13548 (2023).
> >
> > [4] Phillip Keung, Yichao Lu, György Szarvas, and Noah A. Smith. 2020. The Multilingual Amazon Reviews Corpus. In Proceedings of the 2020 Conference on Empirical Methods in Natural Language Processing (EMNLP), pages 4563–4568, Online. Association for Computational Linguistics.

---

> > > ### Author Response · Authors · 2024-12-02
> > > **Reminder for the Discussion**
> > >
> > > Dear Reviewer Uotu,
> > >
> > > We’d like to extend our sincere gratitude for the insightful reviews of our paper again! As we approach the end of the discussion phase, we kindly remind you to discuss any unresolved concerns you have regarding the paper. Your suggestions are invaluable to us in enhancing the manuscript.
> > >
> > > Thank you once again for your patience and contributions!

---

> > > > ### Author Response · Authors · 2024-12-03
> > > > **Last Day Reminder**
> > > >
> > > > Dear Reviewer Uotu,
> > > >
> > > > As today is the last day that reviewers may post a message to the authors, we kindly remind you to discuss any unsolved concerns about our paper. Thanks for your participation again! Your feedback is invaluable in helping us refine and improve our manuscript!

---

### Author Response · Authors · 2024-11-28
**Paper Revision Summary**

We sincerely thank all reviewers for their valuable and constructive feedback on our paper. In response, we have carefully revised the manuscript and would like to take this opportunity to summarize the changes we made. All revisions are highlighted in blue for your convenience.

- **Addressing how GDPO resolves the issue of overlooking minority preferences**:
Reviewer Uotu raised a concern regarding how GDPO addresses minority preferences, which we believe stems from a misunderstanding of the term. To clarify, we have revised Section 3 (Preliminaries) to explicitly define minority preferences as those with the lowest proportion in the belief distribution (Lines 173–177). We further refer the reviewer to Figure 3 of the original submission, which illustrates that GDPO increases reward margins for minority preferences, in contrast to DPO. This demonstrates GDPO's capability to balance and optimize both majority and minority preferences simultaneously. Additionally, Table 3 provides examples of GDPO generating responses that reflect minority beliefs, offering additional evidence of its effectiveness.
- **Generalization of predefined belief sets**:
Reviewers raised concerns regarding the generalizability of GDPO's reliance on predefined belief sets. To address this, we discuss how to conduct the belief mining on other preference datasets in Appendix A. Specifically, using GPT-4o, we extract 23 belief statements from 300 randomly selected examples in HH dataset and estimated a proxy for the belief distribution by aggregating all examples containing these statements. Moreover, in the Introduction (Lines 052–081), we explain how belief distributions can help resolve conflicting preferences in alignment methods. The definition of beliefs has also been updated, with additional references provided in Section 4 (Group Distributional Preference Optimization) on Lines 198–201.
- **Discussion of training efficiency**:
To provide more insights into GDPO's computational performance, we have added a discussion on its training efficiency in Appendix B. In short, GDPO adds a modest computing overhead, measured empirically at 16%, compared to DPO.
- **Extending GDPO beyond DPO**:
Reviewer xrBo suggested evaluating GDPO on methods beyond DPO to enhance its impact. We agree and have started experiments with the KTO method. However, due to time constraints, the updated results could not be included in the current submission. We will share our findings on the forum as soon as they are available.

Once again, we extend our heartfelt thanks to all reviewers for their efforts and insightful comments. We kindly encourage you to review our revised submission and hope that our updates adequately address your concerns. We would greatly appreciate it if you consider revising your scores accordingly.

---

### Author Response · Authors · 2024-12-04
**Rebuttal Summary**

We sincerely thank all reviewers for their insightful comments on our paper. We'd like to take this opportunity to summarize our key contributions and address the main concerns raised during the review process.

**Key Contributions**
- We introduce group distributional preference optimization task, a novel perspective for group-wise alignment that ensures models capture and represent the diversity within a group.
- We identify challenges in current alignment methods, such as DPO, which skew toward dominant preferences during training, making them inadequate for achieving group distributional preference alignment.
- We integrate the concept of human belief from epistemology into pluralistic preference alignment. This innovation addresses DPO’s limitation of overlooking minority preferences, resulting in a more inclusive and representative language model.

**Replies to Key Concerns**
- **Applying "Belief" to General Preference Alignment Tasks**:
The concept of belief is not tailored to specific tasks but is intrinsic to preference alignment, as human preferences are fundamentally shaped by underlying beliefs from an epistemological perspective. We have added a belief mining section to the appendix, focusing on the belief mining of HH dataset using GPT-4o. Human evaluation shows that GPT-4o achieves 83% accuracy in annotating beliefs on HH. It show that belief mining on existing preference alignment datasets is feasible and efficient with the language understanding capabilities of LLMs.
- **Integrating GDPO with Various Alignment Losses**:
Additional experiments integrating GDPO with KTO yield results consistent with those from our DPO-based experiments ([details here](https://openreview.net/forum?id=bgpNJBD6Va&noteId=zjUBFjBNhB)). These experiments demonstrate that GDPO is compatible with paired preference alignment methods (e.g., DPO) and adaptable to unpaired methods (e.g., KTO). This supports our claim that GDPO is a generic framework applicable to a wide range of alignment losses, rather than a mere incremental extension of DPO.

We greatly appreciate the reviewers' valuable feedback, which has helped us refine and strengthen our work! We will include these experimental results and updates in the next version of our paper. Thanks for your patience and contributions again!

---

### Meta-Review · Area_Chair_J7Tf · 2024-12-21

**Metareview:**

This paper identifies the issue in existing consistency optimization methods for large language models (LLMs), which prioritize majority preferences within a group while overlooking minority preferences. To address this problem, the authors propose a novel framework, GDPO, which estimates the distribution of beliefs within a group and uses it as a condition to align preferences, enabling a more inclusive reflection of diversity.

The focus on addressing the important and interesting challenge of considering minority preferences is a significant contribution. The explanations and experiments, which were strengthened through discussions with reviewers, further support the claims of the paper.

While challenges remain, such as the depth of the technical contribution and the reliance on predefined beliefs and the quality of belief generation, the overall ideas presented in this paper are worth sharing with the research community.

**Additional Comments On Reviewer Discussion:**

During the rebuttal phase, one reviewer acknowledged the improvements made to the paper and increased the score to 6.
On the other hand, the responses to the reviewer who initially gave a rejection score (3) appear reasonable and well-justified, but the reviewer did not provide any feedback.
It seems that raising the score to at least 4 or 5, would be appropriate. In that case, the paper would have an enough chance of acceptance.

---

### Decision · Program_Chairs · 2025-01-22

Accept (Poster)